# Host Defense Peptides LL-37 and Lactoferrin Trigger ET Release from Blood-Derived Circulating Monocytes

**DOI:** 10.3390/biomedicines10020469

**Published:** 2022-02-17

**Authors:** Frederic V. Schwäbe, Lotta Happonen, Sofie Ekestubbe, Ariane Neumann

**Affiliations:** Department of Clinical Sciences, Division of Infection Medicine, Lund University, SE-22184 Lund, Sweden; f.schwaebe@gmail.com (F.V.S.); lotta.happonen@med.lu.se (L.H.); sofie.ekestubbe@skane.se (S.E.)

**Keywords:** host defense peptides, monocytes, neutrophils, neutrophil–monocyte interaction, extracellular traps

## Abstract

Neutrophils are commonly regarded as the first line of immune response during infection or in tissue injury-induced inflammation. The rapid influx of these cells results in the release of host defense proteins (HDPs) or formation of neutrophil extracellular traps (NETs). As a second wave during inflammation or infection, circulating monocytes arrive at the site. Earlier studies showed that HDPs LL-37 and Lactoferrin (LTF) activate monocytes while neutrophil elastase facilitates the formation of extracellular traps (ETs) in monocytes. However, the knowledge about the impact of HDPs on monocytes remains sparse. In the present study, we investigated the effect of LL-37 and LTF on blood-derived CD14^+^ monocytes. Both HDPs triggered a significant release of TNFα, nucleosomes, and monocyte ETs. Microscopic analysis indicated that ET formation by LL-37 depends on storage-operated calcium entry (SOCE), mitogen-activated protein kinase (MAPK), and ERK1/2, whereas the LTF-mediated ET release is not affected by any of the here used inhibitors. Quantitative proteomics mass spectrometry analysis of the neutrophil granular content (NGC) revealed a high abundance of Lactoferrin. The stimulation of CD14^+^ monocytes with NGC resulted in a significant secretion of TNFα and nucleosomes, and the formation of monocyte ETs. The findings of this study provide new insight into the complex interaction of HDPs, neutrophils, and monocytes during inflammation.

## 1. Introduction

In response to bacterial invasion or tissue injuries, neutrophils undergo the process of degranulation, releasing a broad range of proteins from distinct granules [1]. Since neutrophils are often seen as initiators of the immune response, it may be hypothesized that their deployed molecules orchestrate subsequent defense mechanisms. Inflammation driven by neutrophils is an underlying factor in various clinical conditions. While usually being considered short-lived cells, neutrophils display an unnormal prolonged life span during chronic inflammation [2]. 

Upon stimulation by neutrophil or endothelial secretion products, e.g., due to a microbial breach [3], circulating monocytes arrive as a second wave of immune cells. Due to their plasticity, monocyte behavior is majorly distinct to the tissue environment [4]. They account for around 10% of all leukocytes [5]. Monocytes are traditionally divided into three sup-populations, dependent on their expression of the surface markers CD14 and CD16 [5]. Based on this, classical monocytes (CD14^+^CD16^−^) are associated with phagocytosis and immune responses and intermediate monocytes (CD14^+^CD16^+^) are connected to cytokine secretion and antigen presentation while non-classical monocytes (CD14^lo^CD16^+^) play a role in complement and adhesion [6]. Changes in the monocyte subsets have been linked to bacterial and viral infections, auto-immune disorders, or chronic inflammation [5]. In addition, different subsets of monocytes are distinct in the secretion of TNFα and IL-6 [7]. While classical monocytes were found to be the most efficient producers of cytokines, non-classical monocytes released the lowest cytokine levels [7]. The stimulus together with the released cytokines might, in turn, give an indication towards which type of monocyte is responding and thus if the cell response is directed towards diminishing or perpetuating an inflammation. Intermediate monocytes are often characterized by their ability to produce proinflammatory cytokines and reactive oxygen species (ROS) upon toll-like receptor (TLR) stimulation [6]. Monocytes have long been solely recognized as macrophage precursor cells, and their role in host defense and immunity, however, remains mostly unclear [3]. 

A correlation between neutrophils and monocyte recruitment to sites of inflammation has been proposed by Janardhan et al. [8]. Infiltration of monocytes mediated by neutrophils has additionally been shown in viral-induced encephalitis [9]. Furthermore, neutrophil-associated host defense peptides LL-37 and LTF have been identified to mobilize inflammatory monocytes [10,11]. LTF triggers the recruitment of macrophages, upregulation of surface markers, and secretion of proinflammatory cytokines from peripheral blood or monocyte-derived dendritic cells [11]. LL-37 can also enhance chemokine expression and, together with another HDP, heparin-binding protein (HBP), plays a direct role in chemoattraction of monocytes [12]. Degranulation of LL-37 and HBP by neutrophils triggers the polarization of macrophages into M1 proinflammatory phenotype [10,13] and adhesion of monocytes to the endothelial tissue [14,15]. In addition, LL-37 induces the formation of neutrophil extracellular traps (NETs; [16]), suggesting that neutrophils may activate their own kind. Stimulation of neutrophils and monocytes leading to ET formation can be acquired by various (bio)chemical inducers, such as phorbol-12-myristate-13-acetate (PMA; [17]) or lipopolysaccharide (LPS) [18], and with a broad range of bacteria, viruses, and parasites [17,19,20,21,22,23]. Recently, it has been demonstrated that host-derived molecules, such as neutrophil elastase and histones, can trigger ET formation in monocytes [6]. In this study, we analyzed the effect of neutrophil-associated HDPs LL-37 and LTF on monocyte extracellular trap formation (ETosis) to shed light on the complex relationship between neutrophils and monocytes. 

## 2. Materials and Methods

### 2.1. Reagents

Phorbol 12-myristate 13-acetate (PMA), 2-Aminoethyl diphenylborinate (2-APB), U0126 ethanolate, SB203580, and Cytochalasin D were all purchased from Merck KGaA (Darmstadt, Germany). LL-37 was purchased from Schafer-N (Copenhagen, Denmark). LTF was purchased from Merck KGaA (Darmstadt, Germany).

### 2.2. Isolating CD14^+^ Monocytes from the PBMC Fraction

CD14^+^ monocytes were isolated by density gradient centrifugation and magnetic bead separation. Here, 10 mL of leukocyte concentrate (pooled, project identification code 2021:10, 21.05.2021, Clinical Immunology and Transfusion Medicine, SUS Lund) were diluted 1:1 with 0.9% sodium chloride (NaCl). In total, 20 mL of the mixture were layered onto 20 mL of Lymphoprep™(Thermo Fisher Scientific, Waltham, MA, USA) and centrifuged for 20 min at 700× *g* without brake. Erythrocytes were lysed with H_2_O for 15 s. Purified PBMCs were resuspended in Magnetic-Activated Cell Sorting (MACS) buffer and CD14 microbeads (both Miltenyi Biotec, Bergisch Gladbach, Germany) were added. Cells were then sorted with an LS column using an MACS separator (both Miltenyi Biotec, Bergisch Gladbach, Germany). After the separation, cells were resuspended in RPMI 1640 (Thermo Fisher Scientific, Waltham, MA, USA), and cell numbers were adjusted to the respective experiments. 

### 2.3. TNFα and Nucleosome Release

For the analysis of the nucleosome release, 1 × 10^6^/mL CD14^+^ cells were incubated with NGC, PMA, or peptides for 3 h. Cells were then spun down for 5 min at 400× *g*, and supernatants were collected and stored at −80 °C. Cell Death Detection ELISAPLUS (Roche Holding, Basel, Switzerland) was used for the analysis of nucleosome release according to the manufacturer’s recommendations. TNFα and IL-6 secretion was analyzed using an ELISA kit from Thermo according to the manufacturer. Absorbance for both assays was measured at 450 nm. 

### 2.4. Oxidative Burst

The production of reactive oxygen species (ROS) was applied to analyze neutrophil activation. In total, 1 × 10^6^/mL monocytes were seeded out in triplicates for each sample. Cells were labeled by adding 100 μL of 2,7 di-chlorofluorescein diacetate (DCF-DA; Thermo Fisher Scientific, Waltham, MA, USA), at a final concentration of 50 μg/mL, and incubating for 20 min at RT. Unlabeled cells served as a control for autofluorescence. After incubation, the plate was centrifuged for 5 min at 370× *g* and RT. The supernatant was removed and 100 μL of stimuli were added to the cells. Fluorescence was measured at an excitation wavelength of 485 nm and emission wavelength of 530 nm using a Victor 3 microplate reader (PerkinElmer Inc., Waltham, MA, USA) at t_0h_ and t_3h_.

### 2.5. Monocyte ET Induction and Immunofluorescence Microscopy

To observe and quantify monocyte extracellular trap (MoET) formation, immunofluorescence microscopy was used, as it was described for neutrophils [24]. Cover slips were placed in 48-well plates and coated with poly-L-lysine (Thermo Fisher Scientific, Waltham, MA, USA) for 20 min. In total, 5 × 10^5^ cells were seeded with 100 μL per well. Then, 100 μL of stimuli were added and the plate was incubated at 37 °C for 3 h. For the inhibition experiments, cells were pre-incubated with the respective inhibitors for 30 min, and then stimuli were added for an additional 3 h of incubation. Samples were fixed with 2% PFA, permeabilized with 1X PBS + 0.5% Triton X-100 for 1 min, and blocked with 1X PBS + 0.05% Tween 20 + 2% BSA for 20 min at RT. Anti-DNA/Histone H1 Antibody (Merck KGaA, Darmstadt, Germany) diluted 1:5000 in blocking solution was added and the plate was incubated for 1 h at RT. Goat anti-Mouse IgG (H + L) Cross-Adsorbed Secondary Antibody, Alexa Fluor 488 (Thermo Fisher Scientific, Waltham, MA, USA) diluted 1:1000 in blocking solution was added. The plate was incubated shielded from light for 1 h at RT. Cells were embedded in ProLong™ Gold Antifade Mountant with DAPI (Thermo Fisher Scientific, Waltham, MA, USA). Samples were visualized using an Eclipse Ti-E inverted microscope system (Nikon, Tokyo, Japan). In total, 4 images were taken per stimulus with a “Plan Apo 40X DIC M N2” objective with a numeric aperture (NA) of 0.9. Images were analyzed using Fiji software version 2.1.0/1.53c [1]. Cells were marked either negative or positive for MoETosis and the percentage of MoET release was calculated. One data point in the graph represents one microscopy image analyzed.

### 2.6. Collection and Analysis of Neutrophil-derived Granulation Content (NGC)

PMNs were isolated as described previously [24]. Briefly, freshly drawn blood from healthy donors was collected in citrate tubes (ethical permit 2008/657 Lund University). Blood was layered on PolymorphPrep (Thermo Fisher Scientific, Waltham, MA, USA), and density gradient centrifugation was performed. Erythrocytes were lysed with endotoxin-free water and cells were finally resuspended in RPMI 1640 (Thermo Fisher Scientific, Waltham, MA, USA). After 4 h of incubation at 37 °C, cells were then mechanically activated by thermal shock. For this, samples were quickly switched between 45 °C and 0 °C to 45 °C, vortexed, and then spun for 20 min at 22,000× *g* to collect proteins. The samples were pooled from several donors and stored at −20 °C in aliquots until further usage. The overall protein content was determined using Pierce 660 nm assay (Thermo Fisher Scientific, Waltham, MA, USA). For SDS-PAGE, 2 µg were mixed (1:1) with reducing loading buffer (Thermo Fisher Scientific, Waltham, MA, USA), denatured at 95 °C for 10 min, and then loaded on a 10–20% Novex Tricine pre-cast gel (Thermo Fisher Scientific, Waltham, MA, USA). Electrophoresis was performed at 120 V for 90 min. The gel was stained using Coomassie Brilliant blue (Thermo Fisher Scientific, Waltham, MA, USA), and images were obtained using a Gel Doc Imager (Bio-Rad Laboratories Inc., Hercules, CA, USA).

### 2.7. Sample Preparation for Mass Spectrometry

For in-solution digestion, 50 µL of the NGC at a protein concentration of 70 µg/mL were denatured with 8 M urea, 100 mM ammonium bicarbonate, 5 mM tris(2-carboxyethyl)phosphine (TCEP) in a final volume of 100 µL at 37 °C, 800 rpm for 60 min in triplicates. The disulfide bonds were reduced with a final concentration of 10 mM iodoacetamide at 22 °C, for 30 min in the dark. The urea was diluted to a concentration of below 1.5 M with the addition of 250 μL of 100 mM ammonium bicarbonate, and the proteins digested with 2 µL of 0.5 µg/µL of sequencing-grade trypsin (Promega, Fitchburg, WI, USA) at 37 °C, 800 rpm for 18 h. The digested samples were acidified with 10% formic acid to a pH of approximately 3.0 and cleaned for mass spectrometry using silica C18 reverse phase MacroSpin columns (Thermo Fisher Scientific, Waltham, MA, USA) according to the manufacturer’s instructions. For in gel digestion, 2 µg of protein were mixed (1:1) with reducing loading buffer (Thermo Fisher Scientific, Waltham, MA, USA), denatured at 95 °C for 10 min, and then loaded on a 10–20% Novex Tricine pre-cast gel (Thermo Fisher Scientific, Waltham, MA, USA). Electrophoresis was performed at 120 V for 90 min. The gel was stained using Coomassie Brilliant blue (Thermo Fisher Scientific, Waltham, MA, USA), and images were obtained using a Gel Doc Imager (Bio-Rad Laboratories, Hercules, MA, USA). The most prominent bands were excised and prepared for mass spectrometry as described [25]. The peptide concentration of all samples was measured on a Nanodrop (Thermo Fisher Scientific, Waltham, MA, USA) and approximately 150 ng of peptides were analyzed by mass spectrometry. 

### 2.8. Liquid Chromatography Tandem Mass Spectrometry (LC-MS/MS)

All peptides were analyzed on an Orbitrap Eclipse mass spectrometer connected to an ultra-high-performance liquid chromatography Dionex Ultra300 system (both Thermo Fisher Scientific, Waltham, MA, USA). The peptides were loaded and concentrated on an Acclaim PepMap 100 C18 precolumn (75 μm × 2 cm) and then separated on an Acclaim PepMap RSLC column (75 μm × 25 cm, nanoViper, C18, 2 μm, 100 Å) (both columns Thermo Fisher Scientific, Waltham, MA, USA), at a column temperature of 45 °C and a maximum pressure of 900 bar. A linear gradient of 3% to 38% of 80% acetonitrile in aqueous 0.1% formic acid was run for 90 min. In total, 1 full MS scan (resolution 120,000; mass range of 350–1400 *m*/*z*) was followed by MS/MS scans (resolution 15,000) of the 20 most abundant ion signals. The precursor ions were isolated with a 1.6 *m*/*z* isolation window and fragmented using higher-energy collisional-induced dissociation (HCD) at a normalized collision energy of 30. The dynamic exclusion was set to 45 s. 

### 2.9. Data Analysis

Acquired MS raw spectra were analyzed using Proteome Discoverer 2.5 (Thermo Fisher Scientific, Waltham, MA, USA) against an in-house compiled dataset containing the reviewed Homo sapiens proteome, UniProt ID UP000005640. Fully tryptic digestion was used allowing 2 missed cleavages. Carbamidomethylation (C) was set to static and protein N-terminal acetylation and oxidation (M) to variable modifications. Mass tolerance for precursor ions was set to 10 ppm, and for fragment ions to 0.02 Da. The protein false discovery rate (FDR) was set to 1%. Proteins quantified by 2 or more unique peptides were considered as relevant and used in R [2] for total ion current (TIC) normalization of the data. The MS non-TIC normalized raw data for the NGC in-solution digest is presented in Appendix A, and that of the SDS-PAGE fractionated gel bands in Appendix A. The mass spectrometry data were deposited to the ProteomeXchange [26] consortium via the MassIVE partner repository (https://massive.ucsd.edu/; accessed on 2 February 2022) with the dataset identifier PXD031375, and are presented in Appendix A.

### 2.10. Statistical Analysis

Data were analyzed by using GraphPad Prism v7.0 (GraphPad Software, San Diego, CA, USA). Differences between 2 groups were analyzed by using a paired, 1-tailed Student *t* test or 1-way ANOVA with the Bonferroni post hoc test. Significance is indicated as * *p* ≤ 0.05, ** *p* ≤ 0.01, *** *p* ≤ 0.001, and **** *p* ≤ 0.0001. ns indicates no statistical significance.

## 3. Results

### 3.1. LL-37 Triggers Secretion of TNFα, Nucleosomes, and ETs from CD14^+^ Monocytes

Neutrophils and monocytes share a complex relationship, orchestrating each other in addition to other immune cells [27]. For the first set of experiments, we thus sought to analyze the effect of LL-37 and LTF on monocytes regarding ROS production, TNF and IL-6 release, and ET formation. Here, we incubated CD14^+^ monocytes with LL-37 and LTF at concentrations described in the literature [28,29]. PMA served as a positive control for the activation of monocytes [17] (Figure 1a–c). To exclude the cytotoxic effects of the stimulation, we measured the LDH release (Appendix A). This analysis showed no detrimental effect on the cells mediated by the used agents. The interaction of peptides with monocytes has been reported to affect cytokine secretion [10,30]. As seen in Figure 1a, the incubation with PMA resulted in a significant secretion of TNFα within 3 h. While the stimulation with LL-37 or LTF was not as efficient compared to PMA, a significant increase in TNFα release was still detected after treatment with LL-37 or LTF treatment (Figure 1a). Analysis of IL-6 secretion revealed that LTF had a significant impact (Figure 1b), whereas no response was detected when cells were treated with PMA or LL-37. Next, we sought to analyze the effect of PMA and the peptides regarding monocyte ET formation. Various studies have reported that PMA is a potent inducer of neutrophil extracellular traps [18,31] and recently also for human CD14^+^ monocytes [17]. Correlating with the findings of Granger and colleagues, in our experimental set up, CD14^+^ monocytes responded to PMA with the secretion of DNA-histone complexes/nucleosomes (Figure 1c). The formation of neutrophil ET in response to 5 µM of LL-37 was reported earlier [16]. Here, we found that 5 µM LL-37 also significantly induced the release of nucleosomes from monocytes within 3 h (Figure 1c). In addition, 75 nM LTF showed significant nucleosome release from CD14^+^ monocytes with only low standard error of the mean (Figure 1c). 

For neutrophils, ROS-dependent and -independent mechanisms of ET formation have been described. PMA or live *Staphylococcus aureus* induce the assembly and activation of NADPH oxidase [32] while depletion of cholesterol using cyclodextrin led to the release of NET fibers, independently of NADPH oxidase [33]. PMA, an activator of NADPH oxidases, has been shown to induce oxidative burst in PBMC-derived monocytes and macrophages [34]. Here, we found PMA significantly triggered an oxidative burst in primary monocytes within 3 h (Figure 1d). Analyzing the production of ROS in response to LL-37 and LTF revealed no detectable increase but rather a quenching of fluorescence in the presence of the proteins (Figure 1d). 

DNA-intercalating dyes, such as SytoxGreen^TM^, as a method for ET quantification are commonly used in the literature [35]. However, we have previously shown that LL-37 quenches the fluorescence of the DNA-intercalating dye PicoGreen^TM^, suggesting a competitive binding of LL-37 to the DNA [36]. We therefore decided to additionally analyze the interaction of monocytes with LL-37 and LTF by immunofluorescence microscopy. ETs were visualized with an antibody against DNA-histone complexes. The representative images shown in Figure 2a reveal that monocytes treated with all three stimuli significantly released extracellular DNA-histone fibers. 

Quantitative analysis of the images further confirmed that this phenotype is significant (Figure 2b). The highest percentage of released ETs was detected when cells were stimulated with LL-37 (Figure 2b). Various studies have demonstrated that neutrophil ETs from different animal species are released in an ERK1/2-, MAPK-, or store-operated calcium signaling (SOCE) -dependent fashion [37,38,39,40]. This variety of mechanisms led us to further investigate the pathways involved in monocyte ET formation. We therefore pre-incubated the monocytes with cytochalasin D (inhibitor of actin rearrangement), 2-ABP (inhibitor of SOCE), SB203580 (inhibitor of MAPK; [41]), and U0126 (inhibitor of ERK1/2). As seen in Figure 3a,b, ET formation mediated by PMA can be completely diminished by blocking of the ERK1/2 pathway. LL-37-mediated ET release was affected by pre-incubation with inhibitors of the intracellular calcium storage (SOCE), MAPK, and ERK pathway (Figure 3a,c). Previously, it was reported that LTF blocks ET release in neutrophils, thus acting as an intrinsic inhibitor of NET formation in circulation [42]. In contrast to these earlier findings, we observed that LTF significantly facilitated the release of ETs from monocytes (Figure 3a,d). Further analysis indicated that none of the used pathway inhibitors (ERK, MAPK, SOCE, actin rearrangement) significantly affected the LTF-mediated ET formation (Figure 3d). These data suggest that different stimuli activate ET formation via different pathways, as previously reported by Kenny and colleagues [43]. 

### 3.2. Neutrophil Granular Content Triggers Monocyte ET Formation

Since we observed ET formation in protein-stimulated monocytes, as a holistic approach, we were interested in the impact of a mixture of neutrophil granular content (NGC) proteins on the monocytes. For these experiments, purified blood-derived neutrophils were subjected to thermal shock to trigger degranulation in the absence of any biological or chemical stimulus. NGC from several donors *(n* = 6) was pooled and analyzed for its impact on monocytes regarding the release of TNFα, IL-6, and nucleosomes. To exclude false positive results, the TNFα, IL-6, and DNA contents of NGC were measured as a background control and the values were subtracted from the values of the monocyte samples. A significant increase in TNFα secretion was observed after incubation of the monocytes with NGC for 3 h (Figure 4a). In addition, a slight increase in IL-6 release was detected (Figure 4b). Moreover, the treatment of monocytes with NGC facilitated increased nucleosome release by two-fold (Figure 4b). While NGC co-incubation triggered the release of proinflammatory cytokines and nucleosomes, no ROS production as an indication of cell activation was observed (Figure 4c). Here, comparable to the results detected with LTF alone (Figure 1d), the fluorescent signal appeared to be quenched by the addition of NGC (Figure 4d). 

To understand the impact of the NGC on the monocytes in detail and to extend our analysis beyond LL-37 and LTF, we analyzed the NGC protein fraction by quantitative proteomics mass spectrometry (MS) (Figure 5, Appendix A). Moreover, the NGC protein fraction was separated by SDS-PAGE, and the five most prominent bands analyzed by MS as above (Figure 5a, Appendix A). In addition to LL-37 and LTF (Figure 5b), we identified 621 proteins in the NGC fraction (Appendix A). Based on the SDS-PAGE fractionation (Figure 5a), the most prominent bands are Lactoferrin, albumin, plastin-2, hemoglobin b, and protein S100-A8. The total ion count for LL-37 and LTF based on the quantitative MS analysis are displayed in Figure 5b. Other HDPs found in the NGC included azurocidin (HBP), myeloperoxidase (MPO), histidine-rich glycoprotein (HRG), and matrix metalloproteinase 9 (MMP-9), which are displayed as a selection in Figure 5c with values derived from the TIC-normalized protein abundances. 

## 4. Discussion

The present study identified a novel role for HDPs LL-37 and LTF, triggering the release of extracellular traps from CD14^+^ monocytes. The release of HDPs from granules is an important process of various cell types in immune defense and intercellular communication. Janardhan et al. previously proposed a correlation between neutrophil-associated HDPs and monocyte recruitment [8], which was later confirmed by Soehnlein et al. [12]. In its role as an HDP, LL-37 contributes to innate immunity as it shows a broad range of antimicrobial activity against bacteria, such as Staphylococcus *(S.) aureus* and Escherichia *(E.) coli* [44]. In addition, it functions as a chemoattractant for monocytes, neutrophils, and other leukocytes, regulating the inflammatory response [45]. Similar to LL-37, LTF serves as a chemoattractant for monocytes [11], but it can also be used as a preventative drug against sepsis in pre-term infants [46]. Moreover, the LTF-derived peptide hLF1–11 displays LPS binding and antimicrobial characteristics while it did not affect host cells at the concentration used in the present study [29]. Autonomous production of TNFα was found to be crucial for monocyte development and survival [47], thus the release of TNFα can be used as marker for the activation of circulating monocytes [48]. In our experiments, we found that LL-37 and LTF significantly elevated the secretion of TNFα from CD14^+^ monocytes. This contrasts with earlier findings reporting that both HDPs reduced the LPS-triggered cytokine release [49,50,51]. Mortality in LPS-challenged mice was dramatically reduced when animals were treated with LTF, which was correlated with reduced TNF levels in murine serum [51]. Nonetheless, in healthy individuals, the regulatory role of LTF results in the spontaneous release of TNF from PBMCs [52]. 

Enhanced release of TNFα has been associated with the generation of neutrophil ETs [53]. Indeed, we found that monocytes released nucleosomes in response to LL-37 and LTF. LL-37 has previously been shown to facilitate NET formation [16] and is also found within NET structures [36]. An earlier study by Okubo and colleagues suggested that LTF acts as an intrinsic inhibitor of ET formation, since application of this exogenous protein resulted in reduced extracellular DNA fibers. However, it must be noted that in this study, LTF was always added together with PMA [42] while in our experiments, LTF was used alone. Interestingly, compared to PMA, neither HDP address ROS-dependent monocyte activation. ROS-independent pathways have been reported before in neutrophils, e.g., NET induction by *S. aureus* [54] and *Leishmania amazonensis* [55]. Depletion of cholesterol has been observed to induce NET formation [33], allowing the hypothesis that certain membrane alterations might also trigger monocyte activation or render the cells more susceptible to certain stimuli. In addition to this, Arai and colleagues found that uric acid induces ETs without activation of NADPH-oxidase with partial involvement of the NF-κB pathway [56]. Furthermore, the MAPK pathway plays a role in neutrophil ET induction when triggered by PMA, *Helicobacter pylori* or, *Neospora caninum* [57,58,59]. This diversity in possible pathways might also be adapted to monocyte ET formation [43]. 

DNA-intercalating dyes are frequently used to quantify ETs [35]. However, as demonstrated by us previously, LL-37 competes with PicoGreen^TM^ for the binding site on DNA [36]. In this study, we hypothesized that a positive signal could be lost due to this competitive interaction. Both LL-37 and LTF are cationic molecules, prone to binding to negatively charged structures. We therefore propose the importance of utilizing additional markers in addition to DNA-intercalating dyes for ET quantification, e.g., histones, neutrophil elastase, or myeloperoxidase. 

Store-operated calcium signaling (SOCE) is important in neutrophil activation, being considered as the main entry of calcium into different immune cells [60]. Various studies have shown the involvement of SOCE in the formation of monocyte extracellular traps [57,59,61]. Interaction studies of LL-37 with ion channels have revealed the importance of this HDP on calcium signaling in neutrophils and monocytes [62,63]. For neutrophils, the impact of LTF on calcium mobilization was described earlier [64]; however, in our experiments using CD14+ monocytes, no significant effect was detected. 

Protein kinases play a crucial role in the formation of extracellular trap release [38,58,65]. In neutrophils, the treatment of mice with SB203580 showed reduced NET production in the lungs and bronchoalveolar lavage fluid (BALF) of poly I:C-challenged animals [66]. In the presented study, we found that LL-37-triggered monocyte ET formation was significantly decreased by the inhibition of p38 MAP kinase. 

PMA is a direct activator of protein kinase C (PKC). Neeli and Radic reported that inhibition of PKC reduced ET formation in neutrophils [67]. PKC in turn activates NADPH oxidases, resulting in the production of ROS in response to PMA. In our experiments, only in PMA-treated monocytes was ROS accumulation observed, leading to the hypothesis that LL-37- and LTF-mediated ET release occurs independently of NADPH oxidases. Activation of PKC by PMA additionally triggers the activation of the Raf-MEK-ERK pathway [58]. Indeed, we showed that in the presence of U0126, PMA- and LL-37- mediated traps are significantly diminished, suggesting that these are dependent on the Raf-MEK-ERK pathway. For both intracellular calcium mobilization and kinase (ERK and p38) activation, the involvement of LL-37 has been demonstrated in human macrophages affecting Leukotriene B4 production [68]. Neither SB203580 nor U0126 (inhibitors of p38 MAPK and Raf-MEK-ERK, respectively) had an effect on LTF-triggered ET formation, even though the involvement of LTF in the ERK signaling pathway has been reported earlier [69]. 

Cytochalasin D is a potent inhibitor of actin filament cytoskeleton rearrangement during ETosis [70]. The application of CytD up to 15 min after the activation of neutrophils leads to a significant reduction in ET formation, as reported by Neubert and colleagues [71]. In good correlation with the findings of Granger et al., the pre-incubation of the monocytes with CytD had no effect on the PMA- (and HDP-) mediated ET formation. Taken together, our findings agree with previous reports that ET formation triggered by different stimuli occur through various signaling pathways [43]. 

The predominant focus on the analysis of ET formation utilizes single molecules. It should be considered, however, that purified or recombinant proteins can have different/opposite effects compared with native proteins or a mix of several proteins. Still, when we analyzed the neutrophil granular content, we observed similar effects on the monocytes compared to LL-37 and LTF regarding TNFα, nucleosome, and ROS secretion. A mixture of proteins, such as, for example, in FBS used for cell culture, affects the degradation of Mg differently compared to the single proteins contained in the FBS [72]. While contamination with LPS or other agents cannot be excluded completely, we hypothesize that it is important to consider a more holistic approach to immune cell stimulation. In this regard, it might also be interesting in the future to investigate how a mixture of immune cells, e.g., in whole blood, might affect the outcome compared to a single type of purified cells. 

Here, we characterized the effects of the previously described HDPs LL-37 and Lactoferrin, the major NGC constituent identified by quantitative MS analysis, on ETosis. This study, moreover, serves as a repository of other NGC proteins, as we identified more than 600 of these (Appendix A). Future studies will extend this data repository to include the protein composition of monocyte ETs triggered by various stimuli, as performed with neutrophils [73,74].

Overall, the results of this study fit the perception of neutrophils being recruiters and activators of monocytes in this complex relationship. The findings of our study suggest that neutrophils’ involvement in immunity reaches even further than hitherto known. The generation of monocyte ETs triggered by neutrophil-associated HDPs could constitute a mechanism of amplifying the immune response at the site of infection. Thus, understanding the pathways and mechanisms behind monocyte ET formation still demands additional research. 

## Figures and Tables

**Figure 1 biomedicines-10-00469-f001:**
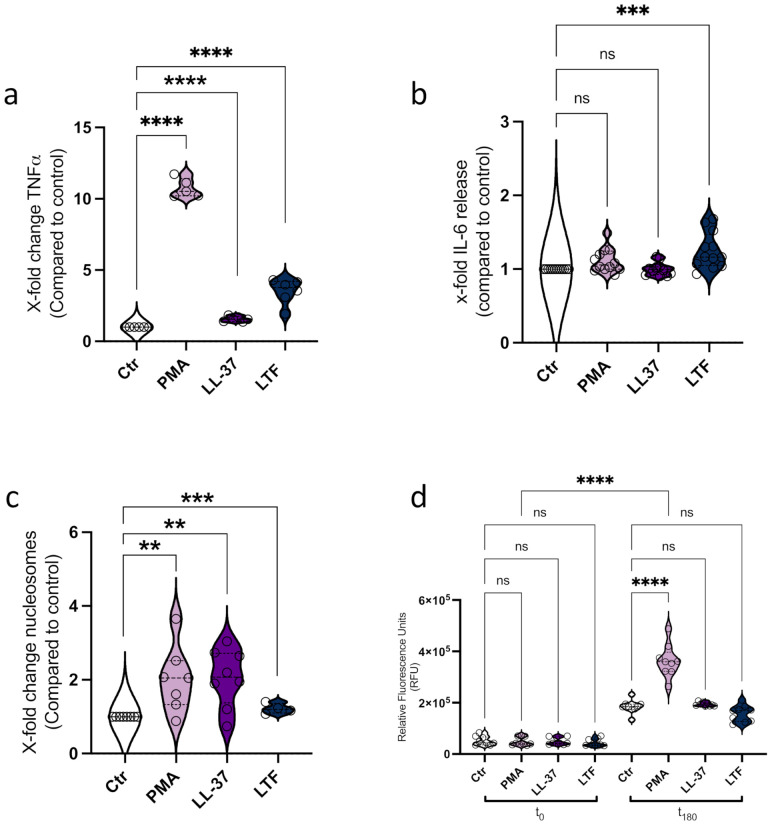
HDPs LL-37 and LTF activate CD14^+^ monocytes. (**a**). Analysis of TNFα secretion in response to 50 nM PMA, 5 µM LL-37, and 75 nM LTF. (**b**). Release of IL-6 after PMA, LL-37, and LTF treatment. (**c**). Nucleosome release/NET formation triggered by PMA, LL-37, and LTF. (**d**). ROS production in response to PMA, LL-37, and LTF measured at t0 and t180. All data represent mean +/– SEM of 3–4 independent experiments. ns = not significant, ** *p* ≤ 0.01 *** *p* ≤ 0.001, **** *p* ≤ 0.0001.

**Figure 2 biomedicines-10-00469-f002:**
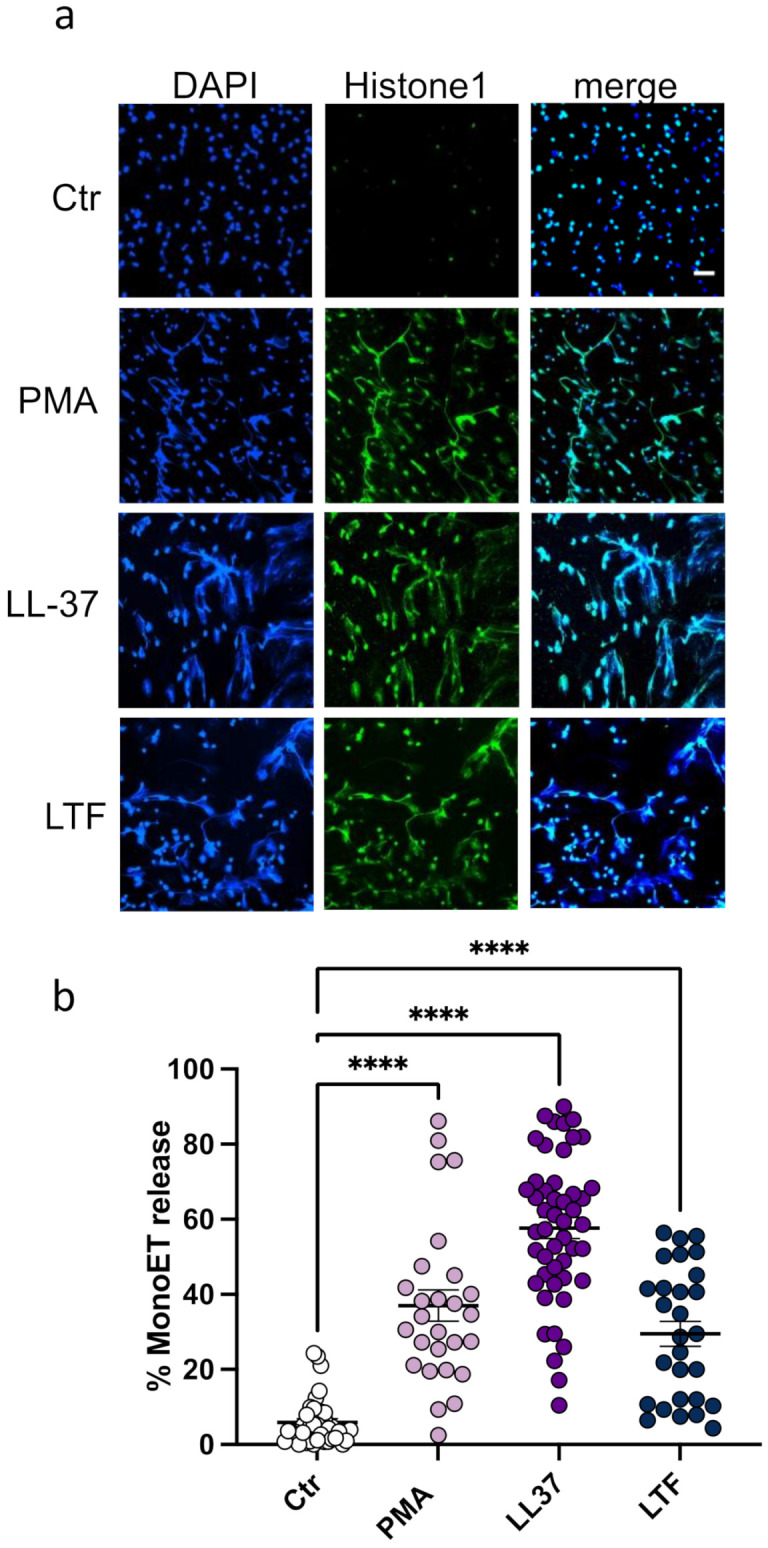
ETs released from CD14+ monocytes in response to LL-37 and LTF. (**a**). Cells were stimulated with different stimuli for 3 h and stained for extracellular DNA (blue) and Histone 1 (green). Scale bar is 50 µm. (**b**). Quantitative analysis of microscopical images shown in (**a**). Every data point represents the relative number of cells undergoing MoET formation in a microscopic image. All data represent mean +/− SEM of 6–11 independent experiments. **** *p* ≤ 0.0001.

**Figure 3 biomedicines-10-00469-f003:**
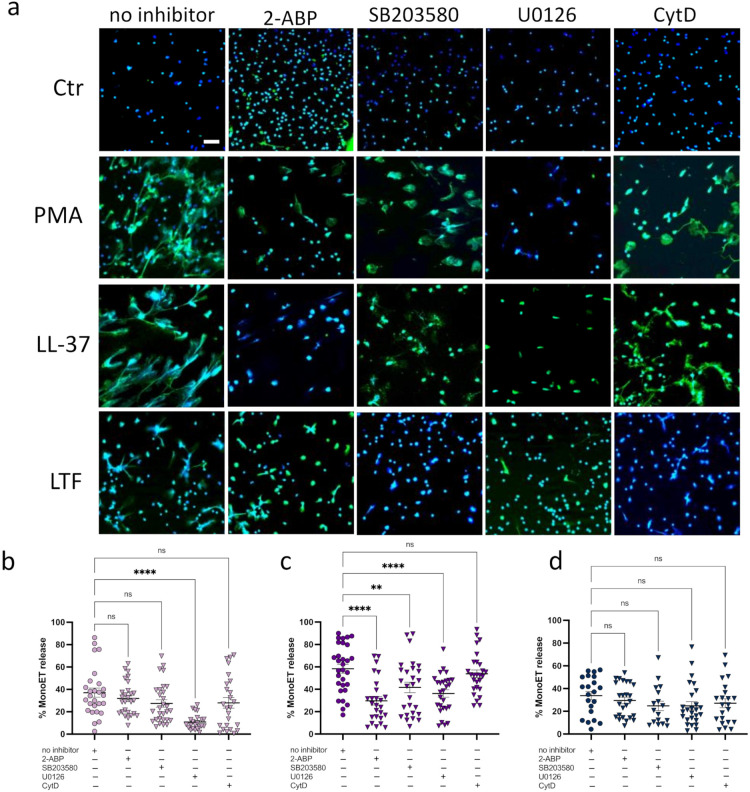
Different stimuli engage various pathways in ET formation. (**a**). Representative fluorescence microscopy images of CD14+ monocytes incubated with PMA, LL-37, or LTF. Blue = DAPI, Green = Histone-DNA complex. Scale bar is 50µm. (**b**–**d**). Quantitative analysis of CD14+ monocytes pre-incubated with inhibitors of different pathways (SOCE, MAPK, ERK1/2, actin rearrangement) and stimulated with (**b**) 25 nM PMA, (**c**) 5 µM LL-37, or (**d**) 75 nM LTF. All data represent mean +/– SEM of 6 independent experiments. ns = not significant, ** *p* ≤ 0.01, **** *p* ≤ 0.0001.

**Figure 4 biomedicines-10-00469-f004:**
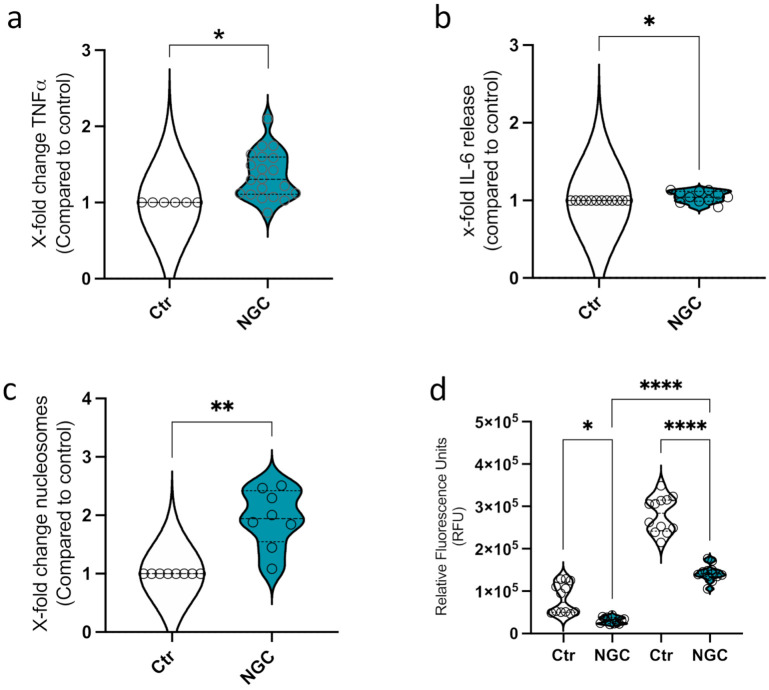
Interaction of primary monocytes with NGC. (**a**). Secretion of TNFα in response to NGC treatment. (**b**). IL-6 release from CD14^+^ monocytes in response to NGC. (**c**). Release of nucleosomes as an indication for ET formation. (**d**). Analysis of ROS production in CD14^+^ monocytes in response to NGC. All data represent mean +/− SEM of 4–5 independent experiments. * *p* ≤ 0.05, ** *p* ≤ 0.01, **** *p* ≤ 0.0001.

**Figure 5 biomedicines-10-00469-f005:**
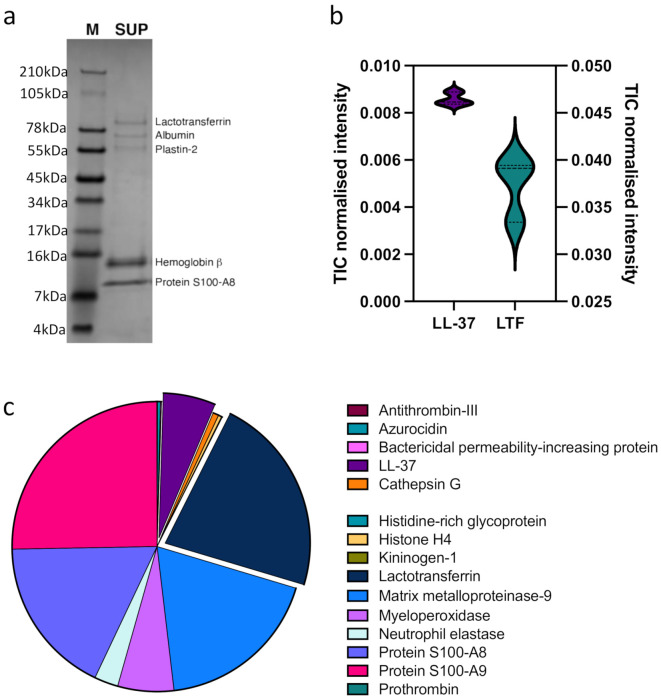
Quantitative MS analysis of NGC. (**a**). SDS PAGE of NGC (=SUP). The most intense gel bands correspond to 1: Lactotransferrin, 2: Albumin, 3: Plastin-2, 4: Hemoglobin β, and 5: Protein S100-A8. (**b**). Total-ion current (TIC) normalized intensity of LL-37 and LTF in NGC. (**c**). Selective representation of various HDPs found in NGC. The proteins are displayed as part of whole, and the size of each protein represents the TIC-normalized abundance. LL-37 and LTF are exploded for easier visualization.

## Data Availability

The mass spectrometry data have been deposited to the ProteomeXchange [26] consortium via the MassIVE partner repository (https://massive.ucsd.edu/ (accessed on 30 December 2021)) with the dataset identifier PXD031375.

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
