# Peer review of "Host Defense Peptides LL-37 and Lactoferrin Trigger ET Release from Blood-Derived Circulating Monocytes"

_biomedicines, 2022, doi:10.3390/biomedicines10020469_

Round 1
Reviewer 1 Report
the study builds upon years of research into monocytes' role in host defense and in inflammation, where they may be keys to either switch off or perpetuate it, and both IL-6 and TNF-a contribute to their decision. This scenario is not clearly delineated in the intro. However the study is carefully conducted and the results are thoroughly described. Some points need to be clarifies in the discussion, e.g. line 377 why uric acid induced ET should be in good correlation with cholesterol depletion? and then, what should the final word be on the ability of HBP to induce ET and nucleosome extrusion, based on negative data in accordance with negative TNF.a induction/release (line 320), or rectifying this for the data in Fig.2 and the discussion on the competitive inhibition of DNA dyes binding for electrochemical reasons? All this should be stated more definitely. Another point for discussion is 1) mention all other HDPs which may be present and be released from neutrophils which have not been investigated, but are-possibly present in NGC hence capable of inducing effects on monocytes and 2) how much lactoferrin is present in Neutrophils, and its respective fractions released outside of the cell, inside cellular compartments including lysosomes, and what reversed for monocyte activation, based on MS estimates. The idea of a NGC repository is good but it should be made available to others, and this is not specified in the text.
Author Response
The study builds upon years of research into monocytes' role in host defense and in inflammation, where they may be keys to either switch off or perpetuate it, and both IL-6 and TNF-a contribute to their decision. This scenario is not clearly delineated in the intro. However, the study is carefully conducted, and the results are thoroughly described.
We thank the reviewer for this positive assessment of our study and appreciate the helpful feedback which clearly improved our manuscript. We have added more information in the introduction to clearly delineate this (page 2, line 58-62).
Some points need to be clarified in the discussion, e.g., line 377 why uric acid induced ET should be in good correlation with cholesterol depletion?
Thank you for this comment. This has been changed to “in addition to this”, since there is no correlation known of (page 11, line 697).
What should the final word be on the ability of HBP to induce ET and nucleosome extrusion, based on negative data in accordance with negative TNFa induction/release (line 320), or rectifying this for the data in Fig.2 and the discussion on the competitive inhibition of DNA dyes binding for electrochemical reasons? All this should be stated more definitely.
We thank the reviewer for this constructive comment. While we think that HBP might play are role in monocyte activation, as previously shown by others, we decided to remove these data entirely from our manuscript and instead introduce Lactoferrin simultaneously with LL-37.
Another point for discussion is 1) mention all other HDPs which may be present and be released from neutrophils which have not been investigated, but are-possibly present in NGC hence capable of inducing effects on monocytes
We agree with the reviewer on this and point to supplemental table 1, where all HDPs found in the NGC are listed. In addition, we created a new figure (Figure 5c), which displays a selection of various HDPs released by the neutrophils in our experimental setup. The selected HDPs are presented as parts of whole derived from the TIC normalized data generated by quantitative MS.
2) how much lactoferrin is present in Neutrophils, and its respective fractions released outside of the cell, inside cellular compartments including lysosomes, and what reversed for monocyte activation, based on MS estimates. The idea of a NGC repository is good, but it should be made available to others, and this is not specified in the text.
We thank the reviewer for this comment. We added the MS data of Lactoferrin released from the neutrophils into figure 5b, displayed as TIC normalization (all metabolites in a sample divided by total number of ions observed in the sample). Moreover, we uploaded the repository to the ProteomeXchange consortium via the MassIVE partner repository (https://massive.ucsd.edu/) with the dataset identifier PXD031375, to make it available for others (page 4, line 230-232). Access information for reviewers; username: MSV000088758_reviewer, passwd: msf**543
Reviewer 2 Report
The study by Schwäbe et al. provides insight into the role that host defence peptides play in the formation of monocyte extracellular traps. By investigating the factors and pathways associated with the better understood process of neutrophil extracellular trap formation, the authors characterised the effect of LL-37 and HBP on monocytes and the formation of ETs. While this manuscript was enjoyable, I have noticed several prominent issues that need to be addressed.
Major concerns
- The microscopy data presented in Figure 2 is not visible and must be repeated for any accurate observations to be made. As much of the proceeding data is based of the ability for PMA, LL-37 and HBP to induce monocyte ET release, it needs to be clearly visible within the microscopy images. For instance, the lack of signal in the PMA condition (which is supposedly the positive control) is concerning. While a slight signal is visible in the LL-37 merge channel, it does not seem to be the result of the combined DAPI and Histone1 signals (which are barely visible). If the authors continue to claim that HBP induces monocyte ET formation, despite the negative result obtained in Figure 1a, the presence of ETs need to be clear in the microscopy images. I am unsure if this issue is due to a technical error during the export of the images, but it appears unlikely that the data generated in Figure 3a could be quantified from the images submitted in Figure 2.
- To support the data presented in Figure 3 b – d, microscopical analysis of monocyte ET formation using the various inhibitors should be performed.
- The sudden shift in focus from LL-37 and HBP towards lactotransferrin alone is jarring and does not flow well with the rest of the study. While removing Figure 5 would address this issue, it may be advantageous to rework the aims of the study and introduce lactotransferrin alongside LL-37 and HBP at the beginning of the study. This way, the data for all three proteins could be presented simultaneously.
- The presence of a strong band attributed to haemoglobin in the SDS PAGE gel (presented in Figure 5a) could indicate contamination with erythrocytes during the neutrophil purification process. Within the literature, there is evidence of haemoglobin expression in nonerythroid cells (Saha et al., 2014, J. Inflamm.) Is there evidence of haemoglobin βexpression within neutrophils which would indicate that the purified neutrophils are not contaminated?
- To support the hypothesis that LL-37 and HBP are key inducers of monocyte ET formation that are released by neutrophils, it would be beneficial to detect these HDPs in the neutrophil granular content. This could be achieved by western blot or immunoprecipitation.
Minor points
- The microscopy data in Figure 2 would benefit from including time lapse images of ET formation induced by the various conditions as opposed to only endpoint images.
Author Response
The study by Schwäbe et al. provides insight into the role that host defence peptides play in the formation of monocyte extracellular traps. By investigating the factors and pathways associated with the better understood process of neutrophil extracellular trap formation, the authors characterised the effect of LL-37 and HBP on monocytes and the formation of ETs. While this manuscript was enjoyable, I have noticed several prominent issues that need to be addressed.
We thank the reviewer for this positive comment. We appreciate the time and effort invested to help us improve our manuscript.
Major concerns
- The microscopy data presented in Figure 2 is not visible and must be repeated for any accurate observations to be made. As much of the proceeding data is based on the ability for PMA, LL-37 and HBP to induce monocyte ET release, it needs to be clearly visible within the microscopy images. For instance, the lack of signal in the PMA condition (which is supposedly the positive control) is concerning. While a slight signal is visible in the LL-37 merge channel, it does not seem to be the result of the combined DAPI and Histone1 signals (which are barely visible). If the authors continue to claim that HBP induces monocyte ET formation, despite the negative result obtained in Figure 1a, the presence of ETs need to be clear in the microscopy images. I am unsure if this issue is due to a technical error during the export of the images, but it appears unlikely that the data generated in Figure 3a could be quantified from the images submitted in Figure 2.
We thank the reviewer for this pertinent comment. We have improved the visibility of the signal in the microscopy images. While PMA is often considered the positive control, it might be possible that other stimuli result in higher ET formation. Nonetheless, we hope that our findings become clearer with improved image quality.
Since HBP is not an antimicrobial peptide, and due to the negative results in figure 1a, we considered to remove all data on HBP and introduce LTF here already (as suggested by the reviewer in a later comment).
- To support the data presented in Figure 3 b – d, microscopical analysis of monocyte ET formation using the various inhibitors should be performed.
We thank the reviewer for this comment. Indeed, all graphs shown in figure 3 are based on microscopy images. We added representative images from our experiments for Ctr, PMA, LL-37 and LTF with and without inhibitors for clarity (see new figure 3a).
- The sudden shift in focus from LL-37 and HBP towards lactotransferrin alone is jarring and does not flow well with the rest of the study. While removing Figure 5 would address this issue, it may be advantageous to rework the aims of the study and introduce lactotransferrin alongside LL-37 and HBP at the beginning of the study. This way, the data for all three proteins could be presented simultaneously.
We agree with the reviewer’s comment, and we think that this helped us to improve our manuscript greatly. We have removed the data on HBP, and instead introduced LTF here. LL-37 and LTF are now presented simultaneously. We think that this feedback improved the flow of the study.
- The presence of a strong band attributed to haemoglobin in the SDS PAGE gel (presented in Figure 5a) could indicate contamination with erythrocytes during the neutrophil purification process. Within the literature, there is evidence of haemoglobin expression in nonerythroid cells (Saha et al., 2014, J. Inflamm.) Is there evidence of haemoglobin βexpression within neutrophils which would indicate that the purified neutrophils are not contaminated?
Indeed, Haemoglobin can be considered as a host defence peptide, upregulated after LPS challenge in vaginal epithelial cells (Saha et al., 2017). Thus, also non-erythroid cells can express Haemoglobin, as pointed out by the reviewer. Whether the found HGB derives from PMNs in our experimental setup cannot be excluded, but it might also be a contamination of RBCs during PMN isolation, since other RBC proteins were found (see supplemental table 1; Stomatin, Spectrin (alpha and beta chain) and Tropomyosin (Bryk et al., 2017).
- To support the hypothesis that LL-37 and HBP are key inducers of monocyte ET formation that are released by neutrophils, it would be beneficial to detect these HDPs in the neutrophil granular content. This could be achieved by western blot or immunoprecipitation.
We thank the reviewer for this comment. We decided to perform quantitative Mass Spectrometry to detect the HDPs in the neutrophil granular content, since we have an expert for this type of analysis within our team of authors (Dr. Lotta Happonen). Therefore, this technique was the most accessible for us to use in these experiments (see figure 5b and supplemental table 1 and 2).
Minor points
- The microscopy data in Figure 2 would benefit from including time lapse images of ET formation induced by the various conditions as opposed to only endpoint images.
We thank the reviewer for this comment. While this is an interesting point to investigate, we think that this is outside the scope of this here presented study.